# Breastfeeding Supportive Services in Baby-Friendly Hospitals Positively Influenced Exclusive Breastfeeding Practice at Hospitalization Discharge and Six Months Postpartum

**DOI:** 10.3390/ijerph182111430

**Published:** 2021-10-30

**Authors:** Lingling Li, Heqing Song, Yu Zhang, Hang Li, Mu Li, Hong Jiang, Yajuan Yang, Ying Wu, Chunyi Gu, Yulian Yu, Xu Qian

**Affiliations:** 1Department of Obstetrics and Gynecology, Shanghai Changzheng Hospital, Shanghai 200003, China; goddess00@163.com (L.L.); wuying19760717@163.com (Y.W.); 2School of Public Health, Fudan University, Shanghai 200032, China; 20211020051@fudan.edu.cn (H.S.); 17211020124@fudan.edu.cn (Y.Z.); 20211020122@fudan.edu.cn (H.L.); guchunyi@fudan.edu.cn (C.G.); xqian@fudan.edu.cn (X.Q.); 3Vital Statistics Department, Songjiang District Center for Disease Control and Prevention, Shanghai 201600, China; 4School of Public Health, University of Sydney, Sydney 2006, Australia; mu.li@sydney.edu.au; 5China Studies Centre, University of Sydney, Sydney 2006, Australia; 6Key Lab of Health Technology Assessment, National Health Commission of the People’s Republic of China, Fudan University, Shanghai 200032, China; 7Nursing Department, Obstetrics and Gynecology Hospital of Fudan University, Shanghai 200011, China; 8Nursing Department, Shanghai Pudong New District People’s Hospital, Shanghai 200032, China; yuyulian1968@163.com

**Keywords:** Baby-Friendly Hospital, exclusive breastfeeding, lactation, hospital practice, Ten Steps to Successful Breastfeeding

## Abstract

Background: Studies have shown that implementing the Ten Steps to Successful Breastfeeding of the Baby-Friendly Hospital Initiative can protect, promote, and support breastfeeding. However, few studies have valuated the quality of breastfeeding supportive services provided by Baby-Friendly Hospitals from the perspective of service users. Methods: This was a hospital-based prospective study, conducted at eight Baby-Friendly Hospitals with a total of 707 pregnant women in Shanghai, China between October 2016 and September 2021. Breastfeeding supportive services during hospitalization were assessed at childbirth discharge using a 12-question questionnaire based on the Chinese “Baby-Friendly Hospital Evaluation Standards”. Women were followed up on six months postpartum. The impact of breastfeeding supportive services during hospitalization on the exclusive breastfeeding at discharge and six months postpartum were assessed. Results: Of the 707 mothers who completed the survey at discharge, 526 were followed up on six months after delivery. The overall exclusive breastfeeding rate among participants was 34.4% at discharge and 52.1% at six months postpartum. Mothers who received better breastfeeding supportive services during hospitalization were more likely to practice exclusive breastfeeding at hospitalization discharge compared with mothers who received poorer services (aOR: 3.00; 95% CI: 2.08, 4.35; *p* < 0.001). Furthermore, they were also more likely to exclusively breastfeed at six months postpartum (aOR: 1.50; 95% CI: 1.03, 2.22; *p* = 0.033). Conclusion: Better breastfeeding supportive services during hospitalization were significantly associated with higher rate of exclusive breastfeeding at discharge and six months postpartum. More effective measures should be adopted to improve the implementation of the breastfeeding supportive services in Baby-Friendly Hospitals to promote exclusive breastfeeding and better maternal and child health.

## 1. Introduction

The World Health Organization (WHO) recommends breast milk as the most appropriate food for infants under six months of age [1]. Breast milk contains all the nutrients that babies need in the first six months of life, so it is an ideal food for infants. Children who were breastfed had lower risk of obesity and less asthma. Furthermore, breastfeeding mothers also benefited from breastfeeding, with lower rates of breast cancer, ovarian cancer, type II diabetes, and postnatal depression [2]. Mothers are recommended to exclusively breastfeed their children in the first six months and continue to breastfeed until their children are at least two years old [1]. However, global breastfeeding (BF) practice during the first six months remains suboptimal. As a study published in the *Lancet* reported, only 37% of children younger than six months of age were exclusively breastfed in low-income and middle-income countries in 2013 [3]. More recent data showed that globally only 42% of newborns were put to mothers’ breasts within the first hour of birth, and only 40% of infants younger than six months of age were exclusively breastfed [4], while the global target of Infant and Young Child Nutrition Targets for 2030 is 70% [5].

In the Chinese National Nutrition Plan (2017–2030) [6], the target of the exclusive breastfeeding (EBF) rate for infants aged between 0 and 6 months was at least 50% by 2020, and this rate shall increase by 10% from 2020 till 2030. However, in 2019, a national BF survey reported that only 11.3% of newborns were breastfed within 1 hr after birth, and 29.2% of babies had never been exclusively breastfed in the first six months [7].

Breastfeeding mothers often encounter various difficulties [8]: they are less likely to achieve successful BF without access to professional guidance and social support [9,10]. Hospitalization after childbirth is regarded as a critical time and opportunity for BF promotion. To improve the quality of BF supportive services and to establish an enabling environment for mothers to achieve better BF practices, the WHO and UNICEF launched the Baby-Friendly Hospital Initiative (BFHI) in delivery hospitals in 1991 [11,12]. Most countries have implemented the BFHI with the purpose of enhancing BF supportive services to mothers and newborns since then [13,14,15]. Global evidence has demonstrated that [16,17] adherence to the BFHI can improve breastfeeding outcomes, including BF initiation and duration, up to one year postpartum.

Baby-Friendly Hospital Initiative accredited hospitals, also called Baby-Friendly Hospitals (BFHs), have been widely established in China [18,19], with a total of 7036 by 2015 [8]. To strengthen the management of BFHs, the Chinese National Health and Family Planning Commission released the “Baby-Friendly Hospital Evaluation Standards (2014 edition)” in 2014 [20]. In response to the standards, provinces across the country have established the leading groups and the technical guidance groups of BFHs, formulated review and assessment plans, and conducted review and assessments of BFHI-accredited hospitals. As one of the largest cosmopolitan cities in China, Shanghai has approximately 120,000 annual births, and there were 74 accredited BFHs approved by Shanghai Municipal Commission by 2017. However, few studies have evaluated the quality of BF supportive services provided by BFHs from the perspective of service users and its impact on exclusive breastfeeding practices. 

This study aimed to assess the BF supportive services provided by BFHs during childbirth hospitalization, and to explore the association between in-hospital breastfeeding supportive services and EBF at the time of discharge as well as six months postpartum. The study also seeks to provide evidence on EBF at discharge and six months after delivery in order to strengthen advocacy for the BFHI and for the evaluation of hospital breastfeeding supportive services in BFHs in China. 

## 2. Materials and Methods

### 2.1. Study Design

This study was a prospective cohort design conducted in eight BFHs in Shanghai: four at the municipal level, including one Maternal and Child Health (MCH) hospital and three general hospitals; and four at the district level, containing three MCH hospitals and one general hospital; between October 2016 and September 2021.

### 2.2. Sample

Women, irrespective of gestational age and delivery mode, were eligible if they had a singleton delivery and voluntarily participated in this study. The exclusion criteria were (1) mothers with diseases that are not suitable for breastfeeding, such as active tuberculosis, HIV, or breast diseases; (2) no intention to be followed at six months postpartum; (3) newborn conditions influencing BF such as a baby with tongue-tie or orofacial clefts.

### 2.3. Data Collection 

Women who consented to participate in the study in one of the eight BFHs were asked to complete a questionnaire of demographic information including age, household registration, education level, intended time to return to work after delivery, alcohol intake, family monthly income per capita, maternity insurance, and family structure. 

The participants were also asked to complete a 12-question questionnaire about the breastfeeding supportive service they received during hospitalization at the time of discharge. The questions about in-hospital BF supportive services were based on the “Baby-Friendly Hospital Evaluation Standards (2014 edition)” [20], and mapped against the Ten Steps to Successful Breastfeeding [21]. Other questions in the questionnaire included breast or nipple pain, breast milk lactation, food intake before BF started, nipple status (‘*normal*’, ‘*flat*’, or ‘*sunk*’), and maternal mood after delivery (‘*calm*’, ‘*anxious*’, or ‘*irritable*’). Childbirth information, such as the mode of delivery and gestational weeks at delivery were extracted from hospital records. Telephone follow-up was carried out six months postpartum to obtain the exclusive breastfeeding practices [22]. The status of exclusive breastfeeding at six months postpartum were obtained by asking mothers the following two questions. “When did you start giving complimentary food to your baby (months)?” “Before the introduction of complementary food, were you feeding your baby with breast milk only?”

### 2.4. Statistical Analysis

Data were analyzed using the Statistical Package for Social Sciences (SPSS) (version 21.0, SPSS Inc., Chicago, IL, USA). Each of the twelve questions was assigned with a zero or a one score: a total score of zero represented providing no supportive services a total score of twelve represented providing the full range of the supportive services. Based on the total score, the services a woman obtained were categorized into the “high” or the “low” score groups, depending on the woman’s score equal to/above or below the mean score. Descriptive statistics were produced to determine the prevalence of EBF at hospital discharge and six months postpartum. One-way analysis of variance/t-test was used to determine differences for continuous outcomes, the Pearson’s chi-square test was used for categorical outcomes, and trend in proportions was tested using Mantel–Haenszel chi-square (χ^2^) tests. Multivariable logistic regression was used for determining the association between BF supportive services in hospitals and the women’s EBF at discharge and six months postpartum. The EBF in this study was defined as the infant having received no other food or liquid, not even water, except for breast milk and drops or syrups consisting of vitamins, mineral supplements or medicine [23,24]. Unadjusted odds ratios (ORs) and adjusted ORs were calculated for assessing the likelihood of EBF at hospital discharge and six months postpartum. All tests were two-sided and statistical significance was defined as *p* < 0.05.

This study was approved by the ethical board of the School of Public Health, Fudan University. All participants provided written informed consent.

## 3. Results

A total of 707 mothers consented to participate in the study and finished the questionnaire about in-hospital BF service at discharge; 526 of them were followed up on six months after the delivery. For the 707 women participating in the survey at hospital discharge, the age range was from 17 to 42 years, with a mean age of 29.8 years. About half of the mothers did not have Shanghai household registration (50.5%). The majority of the mothers were Han ethnicity (97.3%). Nearly 25% of participants were currently unemployed. About 50% of women planned to return to work within six months postpartum. More than 80% of mothers didn’t consume alcohol during pregnancy. Nearly 60% of the participants had maternity insurance. About two thirds (64.4%) of mothers attended the maternity school provided by the hospitals. Over 70% of mothers were primiparous (71.1%). Around 75% of the newborns’ primary caregivers during hospitalization were their fathers and grandparents. The mean score of BF supportive services received by mothers during hospitalization was 8.4, ranging from 2 to 12. (Table 1). For the 526 women who were successfully followed up on six months postpartum, the rate of exclusive breastfeeding at discharge was 33.3% (175/526); and for those who were lost to follow-up at six months postpartum, it was 37.6% (68/181). The chi-square (χ^2^) test showed no statistical differences in demographic characteristics, the rate of EBF at discharge, and the score of BF supportive services between these two groups.

Table 2 presented the status of postnatal BF supportive services provided by BFHs. Only 24.5% of mothers had early initiation of BF and 43.1% of mothers had skin-to-skin contact (SSC) within one hour after birth. Among the 12 BF in-hospital supportive service items, four services achieved high coverage, including rooming-in, providing BF information by doctors, providing BF information by nurses, and no recommendation of infant formula. The chi-square (χ^2^) test showed a significant difference between MCH hospitals and general hospitals on BF supportive services. MCH hospitals consistently performed better in early initiation and SSC within one hour after birth, no bottle use before BF initiation, rooming-in practice, information provision on BF, restriction on the use of breast milk substitute during hospitalization, encouragement of BF on demand, no recommendation of infant formula, BF guidance and human milk as the primary food after birth (*p* < 0.05).

Using the chi-square (χ^2^) test, primiparous mothers, childbirth in MCH hospitals, earlier initiation time, not feeding of liquid or milk formula before BF initiation, and a higher score of BF supportive services during hospitalization were found to be significantly associated with a higher rate of EBF at hospital discharge (Table 1). While mothers who were younger than 30 years old, using vaginal delivery, and a higher score of BF supportive services during hospitalization were found to be significantly associated with a higher rate of EBF at six months postpartum.

After controlling for mother’s age and education, family monthly income per capita, intended time to return to work, alcohol intake, maternity insurance, family structure, receiving specific BF guidance (such as feeding posture and expressing) during pregnancy, mode of delivery, primiparity, the gender of the newborn, newborns’ primary caregiver during hospitalization, maternal nipple status, maternal mood, and admission to infant intensive care, the multivariable binary logistic regression analysis showed that mothers who received better BF supportive services (≥8 scores) during hospitalization were three times more likely to have EBF practice at the time of discharge, compared with mothers who received poorer BF supportive services (<8 scores) during hospitalization (aOR: 3.00; 95% CI: 2.08, 4.35; *p* < 0.001). Furthermore, mothers received better BF supportive services during childbirth hospitalization had significantly higher rates of exclusive breastfeeding at six months postpartum (aOR: 1.50; 95% CI: 1.03, 2.22; *p* = 0.033) (Table 3).

## 4. Discussion

Our study showed that breastfeeding supportive services in the BFHs of Shanghai need to be improved. This was reflected by the low EBF rate at hospital discharge, assessed from the perspective of service users. The implementation of some key BF supportive services was not satisfactory in the hospitals involved in the research, e.g., the low rate of early initiation and skin-to-skin contact within one hour after childbirth. The BF supportive services were better practiced in the MCH than in general hospitals. Mothers who received more comprehensive BF supportive services during hospitalization were more likely to practice EBF at the time of hospital discharge as well as six months postpartum, compared with mothers who received limited hospital support. 

The result of our study showed that the rate of EBF was 34.4% among women discharged from the hospital after childbirth. This result was in keeping with several Chinese studies. A study in a general hospital in Shanghai reported the EBF rate of 34.5% [25], and the rate was 35.5% from a survey in an MCH in Changchun, Jilin Province [26]. However, it was lower than the result of 46.6% of a survey conducted in 32 Chinese cities [27]. Furthermore, the EBF rate at hospital discharge of our study was lower compared with that found in several international studies, such as the EBF rate at hospital discharge of 50% in the USA [28], 67.9% in Italy [29], and 92.3% in Georgia of Western Asia [30]. The results suggest that some BFHs in Shanghai provided inadequate guidance and support on BF practice for new mothers. Active and effective actions are needed to ensure the capacity of health professionals and BF supportive services provisions in BFHs. It is also important to institutionalize regular external monitoring of the implementation and adherence to the BFHI Ten Steps and provide necessary technical assistance to accredited BFHs [31]. Furthermore, the evaluation should also include the service end users’ perspective on the quality of service provision and outcomes.

BFHs aim to give every baby the best start in life by creating a healthcare environment that supports breastfeeding as the norm. Breastfeeding supportive services provided by BFHs help mothers to gain confidence and skills they need to breastfeed exclusively for six months, and to continue to breastfeed with additional complementary foods for 2 years [32]. BFHI has been proven to be very effective in increasing breastfeeding initiation, exclusive breastfeeding, and breastfeeding duration [33]. BF supportive services early in the postpartum period may promote the establishment of exclusive breastfeeding practices for mothers. A study conducted in rural Western Australia suggested that hospital BFHI practices were powerful predictors of mothers’ exclusive breastfeeding [34]. A previous Chinese study also indicated that hospitalization after childbirth might be an ideal time for BF support when professional guidance and reliable information were available [8]. A recent meta-analysis suggested that the effective implementation of Ten Steps to Successful Breastfeeding in BFHI was the most effective intervention for improving BF rates at the health system level [35]. In our study, BF supportive services provided by hospitals during childbirth hospitalization were significantly associated with a higher rate of EBF at hospital discharge, and MCHs performed better than general hospitals. Furthermore, our study found that in-hospital BF supportive services would have a long-term impact on EBF practice since mothers who received better BF supportive services during hospitalization were more likely to have successful EBF at six months postpartum, compared with mothers who received limited hospital supportive services. This result was consistent with a cross-sectional study conducted in Australia that included 977 healthy mothers [36] which demonstrated that receiving adequate support from health professionals after childbirth was associated with better breastfeeding practice at six months. A feasibility and pilot randomized controlled trial in Hong Kong [37] also showed that providing BF knowledge and emotional support for mothers would make them feel more confident with breastfeeding. Since half of the women in this study (50.5%) gave birth in general hospitals, the improvement of BF supportive services quality in general hospitals would have a significant impact on the overall EBF rate of society as a whole. These findings emphasized the necessity of training and supervision on the implementation of BFHI strategies in general hospitals.

Through our study, it was found that the implementation of some key BF supportive services was not being satisfied in the hospitals involved in the research. For instance, our study found the early initiation and SSC within 1 hr after childbirth in these BFHs was extremely low, only 24.5% and 43.1%, respectively. Two previous studies from Shanghai reported similar findings. A study conducted by the Shanghai Center for Women and Children’s Health in 2017 found that the rate of the early sucking was 54.5% and the rate of SSC was 32.9% [38], and in another study involving seven BFHs, the early BF initiation was 54.9% and the rate of SSC was 32.7% [39]. The benefits of immediate SSC and early BF during the first hour after birth have been attested [40,41]. A meta-analysis conducted by the Lancet Breastfeeding Series Group in 2016 showed that the key practices promoted in BFHI had a significant impact on promoting BF in the first hour [42]. However, our findings indicated that the current hospital BF services were far below the requirements of BFHI that state that hospitals should support mothers to breastfeed by encouraging SSC and helping mothers to put their babies on their breast immediately after birth [28]. This was also found in a recent reassessment of a Baby-Friendly Hospital in Ghana report: early breastfeeding initiation was the least met requirement [43]. In addition, our findings showed that other BF supportive services provided by BFHs in this study were also not completely up to the standard guidelines, which suggested that BFHs did not provide sufficient support to promote breastfeeding. For example, 40% and 31.7% participants in our study, from MCH and general hospitals respectively, had not experienced “encouragement of breastfeeding on demand”. The WHO recommended on-demand breastfeeding for the first two years [44]. Demand feeding could meet the nutritional and immunological needs of the infants and help the mothers to maintain adequate milk supplies. A study in Sweden [45] has proved that educating parents on on-demand feeding could greatly reduce scheduled feeding. Thus, implementation of Ten Steps to Successful Breastfeeding in BFHs should further be strengthened to promote short and long term EBF practices.

This study has several strengths: First, our study included both MCH hospitals and general hospitals; so we can gain a better understanding of the situation of BF supportive practice in these hospitals, and identify the specific challenges faced by different types of hospitals. Our findings that BF supportive services in MCHs were performed better than in general hospitals implied that more can be done to improve and increase exclusive breastfeeding rates in general hospitals. Second, this study was conducted at both the municipal and the district levels, which provided further insight into the strengths and weaknesses from the health system perspective. The participants came from both municipal and district levels, and MCH and general hospitals, which reduced the selection bias of research findings. Third, evaluation of BFHs performance was usually conducted by members of the BFHI leading groups and the technical groups. This study assessed the quality of BF supportive services in BFHs during childbirth hospitalization by determining mothers’ EBF practice at both discharge time and six months postpartum, and their associations with the BF supportive services received in hospitals.

The limitation of our study lies in that although the follow up rate was approximately 75% (526/707), there might be selection bias due to the loss of follow-up between hospital discharge and six months postpartum, even if there were no statistical differences in demographic characteristics between those who were lost to follow-up and those were successfully followed up on. Moreover, we did not include some key factors influencing EBF at six months postpartum for analysis, such as social and family support. Thus, the impact of in-hospital BF supportive services might be biased.

## 5. Conclusions

In conclusion, better BF supportive services during childbirth hospitalization were significantly associated with a higher rate of EBF at discharge and six months postpartum. The implementation of the BFHI’s Ten Steps to Successful Breastfeeding should be strengthened to improve the BF supportive services in BFHs and to support optimal infant feeding practices and promote maternal and child health.

## Figures and Tables

**Table 1 ijerph-18-11430-t001:** Characteristics of participants and the associations with exclusive breastfeeding (EBF) at hospital discharge (*n* = 707).

Characteristic	*n* (%)	EBF at Hospital Discharge		*χ^2^*	*p* Value
		Yes, *n* (%)	No, *n* (%)		
Age (years)					
≤30	435 (61.5)	142 (32.6)	293 (67.4)	1.49	0.22
>30	272 (38.5)	101 (37.1)	171 (62.9)		
Education level					
Junior middle school	104 (14.7)	34 (32.7)	70 (67.3)	0.19	0.91
Senior middle school	127 (18.0)	45 (35.4)	82 (64.6)		
College and above	476 (67.3)	164 (34.5)	312 (65.5)		
Intended time to return to work					
≤6 months	350 (49.5)	115 (32.9)	235 (67.1)	0.70	0.43
>6 months	357 (50.5)	128 (35.9)	229 (64.1)		
Alcohol intake during pregnancy					
No	578 (81.8)	199 (34.4)	379 (65.6)	0.01	1.00
Seldom	129 (18.2)	44 (34.1)	85 (65.9)		
Family monthly income per capita					
≤10,000 RMB	495 (70.0)	171 (34.5)	324 (65.5)	0.02	0.93
>10,000 RMB	212 (30.0)	72 (34.0)	140 (66.0)		
Maternity insurance					
No	253 (35.8)	77 (30.4)	176 (69.6)	2.88	0.24
Yes	418 (59.1)	154 (36.8)	264 (63.2)		
Others	36 (5.1)	12 (33.3)	24 (66.7)		
Family structure					
Nuclear family	350 (49.5)	115 (32.9)	235 (67.1)	0.70	0.43
Non-nuclear family	357 (50.5)	128 (35.9)	229 (64.1)		
Receiving specific breastfeeding guidance (such as feeding posture and expressing) during pregnancy					
Yes	353 (49.9)	122 (34.6)	231 (65.4)	0.01	0.92
No	354 (50.1)	121 (34.2)	233 (65.8)		
Mode of delivery					
Vaginal delivery	331 (46.8)	112 (33.8)	219 (66.2)	0.08	0.96
Forceps delivery	11 (1.6)	4 (36.4)	7 (63.6)		
Cesarean section delivery	363 (51.3)	126 (34.7)	237 (65.3)		
Primiparity					
Yes	502 (71.3)	159 (31.7)	343 (68.3)	5.09	0.03
No	202 (28.7)	82 (40.6)	120 (59.4)		
Newborn sex					
Boy	383 (54.2)	134 (35.0)	249 (65.0)	0.14	0.75
Girl	324 (45.8)	109 (33.6)	215 (66.4)		
Primary caregiver during hospitalization					
Husband	166 (23.5)	60 (36.1)	106 (63.9)	0.76	0.68
Husband and grandparents	532 (75.2)	179 (33.6)	353 (66.4)		
Yuesao ^#^ or others	9 (1.3)	4 (44.4)	5 (55.6)		
Types of delivery hospital					
Maternal and Child Health (MCH) hospital	350 (49.5)	140 (40.0)	210 (60.0)	97.38	0.002
General hospital	357 (50.5)	103 (28.9)	254 (71.1)		
Breast or nipple pain					
Yes	180 (25.4)	71 (39.4)	109 (60.6)	2.76	0.10
No	527 (74.5)	172 (32.6)	355 (67.4)		
Time to start lactation					
On the day of delivery	210 (29.7)	122 (58.1)	88 (41.9)	85.37	<0.001
First day after birth	103 (14.6)	32 (31.1)	71 (68.9)		
Second day after birth	76 (10.7)	26 (34.2)	50 (65.8)		
Third day after birth	104 (14.7)	27 (26.0)	77 (74.0)		
Food feeding before breastfeeding					
No food	221 (31.4)	128 (57.9)	93 (42.1)	79.03	<0.001
Milk powder and other food	482 (68.6)	115 (23.7)	371 (76.3)		
Nipple status					
Normal	612 (86.6)	219 (35.8)	393 (64.2)	4.27	0.12
Flat	59 (8.3)	16 (27.1)	43 (72.9)		
Sunk	36 (5.1)	8 (22.2)	28 (77.8)		
Maternal mood after delivery					
Calm	664 (93.9)	229 (34.5)	435 (65.5)	0.81	0.67
Anxious	30 (4.2)	11 (36.7)	19 (63.3)		
Irritable	13 (1.8)	3 (23.1)	10 (76.9)		
Admission to infant intensive care					
No	577 (81.6)	189 (32.8)	388 (67.2)	3.63	0.06
Yes	130 (18.4)	54 (41.5)	76 (58.5)		
Score of breastfeeding supportive services received during hospitalization					
<8 scores	271 (38.3)	57 (21.0)	214 (79.0)	34.65	<0.001
≥8 scores	436 (61.7)	186 (42.7)	250 (57.3)		

#: a person being employed to take care of the mother and newborn in the first month postpartum.

**Table 2 ijerph-18-11430-t002:** The twelve breastfeeding (BF) supportive services provided by Maternal and Child Health (MCH) Hospitals and general hospitals.

Items	MCH Hospital	General Hospital	Total, *n* (%)	*χ^2^*	*p* Value
1. Early sucking within 1 hr after delivery					
Yes	105 (30.0)	68 (19.0)	173 (24.5)	11.47	0.001
No	245 (70.0)	289 (81.0)	534 (75.5)		
2. Skin-to-skin contact (SSC) within 1 hr after delivery					
Yes	190 (54.3)	115 (32.2)	305 (43.1)	35.10	<0.001
No	160 (45.7)	242 (67.8)	402 (56.9)		
3. Limitation of bottle use before BF initiation					
Yes	302 (86.3)	244 (68.3)	546 (77.2)	32.33	<0.001
No	48 (13.7)	113 (31.7)	161 (22.8)		
4. Rooming-in					
Yes	332 (94.9)	322 (90.2)	654 (92.5)	5.53	0.02
No	18 (5.1)	35 (9.8)	53 (7.5)		
5. Information provision about BF by doctors					
Yes	321 (91.7)	325 (91.0)	646 (91.4)	0.10	0.79
No	29 (8.3)	32 (9.0)	61 (8.6)		
6. Information provision about BF by nurses					
Yes	333 (95.1)	324 (90.8)	657 (92.9)	5.17	0.03
No	17 (4.9)	33 (9.2)	50 (7.1)		
7. Advice on no use of bottle by health staff					
Yes	213 (60.9)	192 (53.8)	405 (57.3)	3.62	0.58
No	137 (39.1)	165 (46.2)	302 (42.7)		
8. No use of breast milk substitute during hospitalization					
Yes	199 (56.9)	174 (48.7)	373 (52.8)	4.67	0.04
No	151 (43.1)	183 (51.3)	334 (47.2)		
9. No food or fluids other than breast milk, unless medically indicated					
Yes	253 (72.3)	108 (30.3)	361 (51.1)	124.95	<0.001
No	97 (27.7)	249 (69.7)	346 (48.9)		
10. No recommendation of infant formula					
Yes	347 (99.1)	333 (93.3)	680 (96.2)	16.55	<0.001
No	3 (0.9)	24 (6.7)	27 (3.8)		
11. Encouragement of breastfeeding on demand					
Yes	210 (60)	243 (68.3)	453 (64.2)	5.24	0.02
No	140 (40)	113 (31.7)	253 (35.8)		
12. Specific BF guidance (such as feeding posture and expressing)					
Yes	288 (82.3)	237 (66.4)	525 (74.3)	12.37	<0.001
No	62 (17.7)	120 (33.6)	182 (25.7)		
Score of 12 BF supportive services during hospitalization					
<8 scores	98 (36.2)	173 (63.8)	271 (38.3)	31.30	<0.001
≥8 scores	252 (57.8)	184 (42.2)	436 (61.7)		

**Table 3 ijerph-18-11430-t003:** Associations between in-hospital BF supportive service and exclusive breastfeeding (EBF) at six months postpartum (*n* = 526).

Variable	EBF at Six Months Postpartum		Crude OR	Adjusted OR	95% CI	*p* Value
	Yes, *n* (%)	No, *n* (%)				
Age (years)						
≤30	187 (56.2)	146 (43.8)	1	1		
>30	87 (45.1)	106 (54.9)	0.64	0.70	0.45–1.07	0.10
Education level						
Junior middle school	35 (47.3)	39 (52.7)	1	1		
Senior middle school	50 (53.8)	43 (46.2)	1.30	1.19	0.62–2.30	0.60
College and above	189 (52.6)	170 (47.4)	1.24	1.25	0.67–2.31	0.48
Intended time to return to work						
≤6 months	129 (32.9)	137 (51.5)	1	1		
>6 months	145 (35.9)	115 (44.2)	1.34	1.19	0.83–1.72	0.35
Alcohol intake during pregnancy						
No	217 (50.5)	213 (49.5)	1	1		
Seldom	57 (59.4)	39 (40.6)	1.43	1.55	0.97–2.49	0.07
Family monthly income per capita						
≤10,000 RMB	196 (52.7)	176 (47.3)	1	1		
>10,000 RMB	78 (50.6)	76 (49.4)	0.67	1.03	0.68–1.57	0.90
Maternity insurance						
No	98 (51.3)	93 (48.7)	1	1		
Yes	160 (52.1)	147 (47.9)	0.86	1.14	0.74–1.77	0.55
Others	16 (57.1)	12 (42.9)	0.56	1.32	0.56–3.08	0.52
Family structure						
Nuclear family	129 (48.5)	137 (51.5)	1	1		
Non-nuclear family	145 (55.8)	115 (44.2)	1.34	1.19	0.83–1.72	0.35
Receiving specific breastfeeding guidance (such as feeding posture and expressing) during pregnancy						
Yes	129 (49.2)	133 (50.8)	1	1		
No	145 (54.9)	119 (45.1)	1.27	1.24	0.87–1.77	0.26
Mode of delivery						
Vaginal delivery	153 (58.4)	109 (41.6)	1	1		
Forceps delivery	5 (50.0)	5 (50.0)	0.71	0.84	0.23–3.07	0.79
Cesarean section delivery	115 (45.5)	138 (64.5)	0.59	0.68	0.47–0.98	0.04
Primiparity						
Yes	197 (31.7)	176 (68.3)	1	1		
No	76 (40.6)	74 (59.4)	0.92	1.27	0.81–2.00	0.30
Newborn sex						
Boy	141 (49.3)	145 (50.7)	1	1		
Girl	133 (55.4)	107 (44.6)	1.28	1.26	0.88–1.82	0.21
Primary caregiver during hospitalization						
Husband	59 (48.8)	62 (51.2)	1	1		
Husband and grandparents	190 (47.6)	209 (52.4)	1.05	1.27	0.85–1.93	0.24
Yuesao ^#^ or others	3 (50.0)	3 (50.0)	0.95	1.16	0.67–2.00	0.60
Nipple status						
Normal	234 (52.0)	216 (48.0)	1	1		
Flat	25 (50.0)	25 (50.0)	0.92	0.90	0.48–1.66	0.73
Sunk	15 (57.7)	11 (42.3)	1.26	1.60	0.68–3.66	0.29
Maternal mood after delivery						
Calm	260 (52.6)	234 (47.4)	1	1		
Anxious	9 (39.1)	14 (60.9)	0.58	0.56	0.22–1.40	0.22
Irritable	5 (55.6)	4 (44.4)	1.13	1.36	0.34–5.53	0.67
Admission to infant intensive care						
No	232 (53.0)	206 (47.0)	1	1		
Yes	42 (47.7)	46 (52.3)	0.81	0.92	0.57–1.48	0.72
Score of breastfeeding supportive services received during hospitalization						
<8 scores	86 (44.6)	107 (55.4)	1	1		
≥8 scores	188 (56.5)	145 (43.5)	1.61	1.51	1.03–2.22	0.03

#: a person being employed to take care of the mother and newborn in the first month postpartum.

## Data Availability

The data presented in this study are available on request from the corresponding author (H.J.). The data are not publicly available due to privacy.

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
