# Peer review of "Breastfeeding Supportive Services in Baby-Friendly Hospitals Positively Influenced Exclusive Breastfeeding Practice at Hospitalization Discharge and Six Months Postpartum"

_ijerph, 2021, doi:10.3390/ijerph182111430_

Round 1

Reviewer 1 Report

Overall an interesting paper reflecting Baby Friendly Hospitals.  It would be useful to include:

1) the differences between the original and lost to follow-up group.  Were they the same demographically and did they have different baby friendly experiences in the hospital..

2) Which of the 12 practices were most important? The study compared MCH and General Hospitals and showed some significant difference between them.  It would be very important to further analyze this data to determine which supportive services would have the biggest impact on promoting exclusivity.

Reviewer 2 Report

Thank you for the opportunity to review this manuscript. This study aimed to assess breastfeeding supportive services provided during childbirth in Baby Friendly Hospitals, exploring the association between services and exclusive breastfeeding at both discharge and six months post-partum. I commend the authors on this work, the manuscript is well structured and this work is invaluable moving forward - evaluation of Baby Friendly Hospitals is essential to ensure that they are achieving what they set out to achieve, and this study provides good evidence to support the importance of these services. I recommend this manuscript for publication, with minor reviews. I have the following comments and questions that should be addressed, although I understand some may not be possible given the study structure/ aims:

Overall:

My main concern is that if the exclusive breastfeeding rate at discharge was 34.4%, then the rate of 52.1% at six months post-partum isn’t truly the exclusive breastfeeding rate (i.e. they were EBF at six months, but may have been mixed fed at or since birth). This needs to be better clarified. Furthermore, how did the authors take other breastfeeding support into account (i.e. support since hospital)?

Further to this, it is likely more appropriate to separate the breastfeeding groups into ‘no BF’, ‘mixed feeding’ and ‘EBF’ – although I understand this was not the aim of the study. This should be at least addressed in the discussion.

Title:

The manuscript would benefit from a title that reflects the findings of the study.

Keywords:

Adding ‘Lactation’ may help you reach additional readers.

Introduction:

Would benefit from a sentence or two on the importance of breastfeeding for both mother and infant (and therefore BFH) - would strengthen the importance of this research.

Materials and Methods:

Can the questionnaire be included as Supplementary material? Were questions exactly as presented in table 2? Readers may be particularly interested in wording of questions. Alternatively, could add more of these details to ‘Data collection’ section.

Were there any participant exclusion criteria? If so, please include in manuscript.

Results:

Was 526/707 the expected retention for this study? You described this in the limitations of the discussion, can you provide any further data as to the participants that were not retained? Eg. Rate of exclusive breastfeeding at discharge, infant illness, pregnancy complications, other reasons that may have contributed to these having poorer breastfeeding outcomes and therefore the study overestimates exclusive breastfeeding outcomes. If this data is available, the manuscript would benefit from addressing it.

Did the BF supportive services scores relate differently to BF outcome depending on which services the score came from? I.e. should different services be weighted differently?

Readers may be interested in a figure showing the distribution of supportive services scores.

Discussion:

The discussion would benefit from specific details on the importance and aims of BFH. Evidence on the importance of breastfeeding for both mother and infant would strengthen the importance of this manuscript.

The discussion would benefit from additional details pertaining to services being poorly implemented, in addition to the early initiation/SSC discussed (Line 242). Eg. 40% and 31.7% participants not experiencing ‘encouragement of breastfeed on demand’ seems worryingly high.   
